# Pleural Effusions on MRI in Autosomal Dominant Polycystic Kidney Disease

**DOI:** 10.3390/jcm12010386

**Published:** 2023-01-03

**Authors:** Jin Liu, Xiaorui Yin, Hreedi Dev, Xianfu Luo, Jon D. Blumenfeld, Hanna Rennert, Martin R. Prince

**Affiliations:** 1Department of Radiology, Weill Cornell Medicine, New York, NY 10065, USA; 2Department of Cardiothoracic Surgery, The Second Affiliated Hospital of Nanjing Medical University, Nanjing 210011, China; 3Department of Medicine, Weill Cornell Medicine, New York, NY 10065, USA; 4The Rogosin Institute, New York, NY 10065, USA; 5Department of Pathology, Weill Cornell Medicine, New York, NY 10065, USA; 6Columbia College of Physicians and Surgeons, New York, NY 10027, USA

**Keywords:** age, ADPKD, pleural fluid, MRI, T2

## Abstract

Autosomal dominant polycystic kidney disease (ADPKD) has cystic fluid accumulations in the kidneys, liver, pancreas, arachnoid spaces as well as non-cystic fluid accumulations including pericardial effusions, dural ectasia and free fluid in the male pelvis. Here, we investigate the possible association of ADPKD with pleural effusion. ADPKD subjects (*n* = 268) and age-gender matched controls without ADPKD (*n* = 268) undergoing body magnetic resonance imaging from mid-thorax down into the pelvis were independently evaluated for pleural effusion by 3 blinded expert observers. Subjects with conditions associated with pleural effusion were excluded from both populations. Clinical and laboratory data as well as kidney, liver and spleen volume, pleural fluid volume, free pelvic fluid and polycystic kidney disease genotype were evaluated. Pleural effusions were observed in 56 of 268 (21%) ADPKD subjects compared with 21 of 268 (8%) in controls (*p* < 0.0001). In a subpopulation controlling for renal function by matching estimated glomerular filtration rate (eGFR), 28 of 110 (25%) ADPKD subjects had pleural effusions compared to 5 of 110 (5%) controls (*p* < 0.001). Pleural effusions in ADPKD subjects were more prevalent in females (37/141; 26%) than males (19/127,15%; *p* = 0.02) and in males were weakly correlated with the presence of free pelvic fluid (r = 0.24, *p* = 0.02). ADPKD subjects with pleural effusions were younger (48 ± 14 years old vs. 43 ± 14 years old) and weighed less (77 vs. 70 kg; *p* ≤ 0.02) than those without pleural effusions. For ADPKD subjects with pleural effusions, the mean volume of fluid layering dependently in the posterior–inferior thorax was 19 mL and was not considered to be clinically significant. Pleural effusion is associated with ADPKD, but its role in the pathogenesis of ADPKD requires further evaluation.

## 1. Introduction

Autosomal dominant polycystic kidney disease (ADPKD) is the most common inherited kidney disease [1]. It is characterized by cysts containing a wide range of fluid volumes in the kidneys, liver, pancreas, prostate and arachnoid spaces [2]. Non-cystic fluid accumulation occurs as pericardial effusions, dilated cisterna chyli, dural ectasia, seminal megavesicles, and free fluid in the male pelvis [3,4,5,6,7,8]. These disease phenotypes are primarily the result of mutations in either the *PKD1* gene or *PKD2* gene, which encode polycystin 1 and polycystin 2, respectively. These proteins are located on primary cilia, and are important for ciliary function [9].

Bronchiectasis and increased risk for nosocomial infections have been associated with ADPKD [10,11]. Cystic fibrosis, which is also characterized by bronchiectasis, is caused by loss of function mutations in the cystic fibrosis transmembrane conductance regulator (CFTR) protein. Chloride transport by CFTR in renal cyst epithelium contributes to cyst fluid accumulation in ADPKD; inhibition of CFTR attenuates this process [12]. However, the specific mechanisms responsible for the pulmonary manifestations of ADPKD have not been defined. As part of an ADPKD research repository, we have been reviewing abdominal magnetic resonance imaging (MRI) biennially with T2-weighted imaging from mid-thorax down through the abdomen and pelvis. We noticed unexpected, asymptomatic small pleural effusions in many ADPKD subjects. In this study we evaluated the prevalence of pleural fluid in ADPKD subjects compared to an age- and gender-matched control population without ADPKD to determine if they are, in fact, a component of the ADPKD disease phenotype.

## 2. Materials and Methods

### 2.1. Patients

This health insurance portability and accountability act (HIPAA) compliant study of existing patient data and images was approved by the local Institutional Review Board (IRB). All ADPKD subjects enrolled in the Rogosin ADPKD Repository provided written informed consent and underwent MR imaging as outpatients. Retrospective review of existing data on control patients was also IRB approved; the requirement for informed consent was waived.

Inclusion criteria for ADPKD subjects were (1) diagnosis of ADPKD based upon the Pei criteria and (2) abdominal MRI with axial T2-weighted images covering from mid thorax down to below the kidneys [13]. The exclusion criteria were (1) laboratory data not available within 12 months of the MRI; (2) patients on dialysis or post-kidney transplantation; (3) heart failure, pneumonia or other pulmonary disease that can cause pleural effusion; (4) surgery within 3 months; (5) pregnancy; (6) active malignancy.

Age, gender, and renal function matched control subjects without ADPKD undergoing abdominal MRI were identified from the electronic medical records, including Epic and picture archiving computer system (PACS). Patients with risk factors for pleural effusion were excluded from the control population, including pneumonia, thoracic tumor, heart failure or surgery within 3 months. Renal function matching was only feasible in a subset of the subjects and was performed based upon the chronic kidney disease (CKD) stage. A subset of these patients with cine images, including the heart, was reported previously [5].

### 2.2. Extraction of Data

Patient demographic information and laboratory data were extracted from the electronic medical records for the date closest to the date of MRI. Kidney, liver and spleen volumes were extracted from the MRI reports prepared prospectively at the time of imaging. Genotype (presence ± of *PKD1* or *PKD2* mutation) was extracted from the Rogosin ADPKD Research Repository database. Cystatin C was not measured in this study.

### 2.3. Image Acquisition

MRI exams were obtained at 1.5T using a body array coil (Signa HDXT, GE Healthcare or Magnetom Aera, Siemens Healthineers) using the parameters shown in Table 1. Pulse sequences included, coronal and axial T2-weighted single shot fast spin echo (SSFSE), 3D spoiled gradient recalled echo T1-weighted images with fat suppression or Dixon fat-water separation and diffusion weighted imaging (DWI) with B = 0, B = 500 and B = 1000 weighted acquisitions.

### 2.4. Image Analysis

Axial T2 and DWI B_0_ MR images were analyzed by 3 independent observers (XY, XZ, MRP) blinded to all patient information. These observers had 6, 10 and 30 years of experience interpreting body MRI. Pleural effusion was designated as present when there was sufficient dependent fluid in the posterior-inferior thoracic pleural space to create a T2 bright fluid band, distinct from sub-pleural fat, at least 2 mm thick over a distance of at least 3 cm on at least two consecutive T2-weighted images. Pleural effusion was quantified using manual contouring to annotate dependent pleural fluid on every axial image using ITK-SNAP software, version 3.8.0 (downloaded 12 June 2019) (Figure 1).

### 2.5. Descriptive Statistics

Mean and standard deviation (SD) were reported for normally distributed continuous variables. For non-normal distributions, determined by Shapiro–Wilk test, variables were reported as median and interquartile range. Frequency and percentage were calculated for categorical variables.

For two-group matched continuous variables, *t*-test was used to assess the significance of normally distributed variables. The Mann–Whitney test was used to assess the statistical significance of non-normally distributed continuous variables. For multi-group continuous variables, analysis of variance was used to assess the statistical significance. For categorical variables, the Chi-squared test was used to assess statistical significance. Inter-observer and intra-observer agreement for identifying pleural effusion was assessed using kappa statistic. For measuring pleural fluid volume on MRI, inter-observer and intra-observer agreement was assessed by interclass correlation coefficient (ICC).

### 2.6. Regression Models

Bivariate analysis was used to assess the correlation between the presence of pleural effusion and gender, age, height, weight, body mass index (BMI), blood pressure (BP), liver volume, total kidney volume, spleen volume, free pelvic fluid, aspartate aminotransferase (AST), alanine transaminase (ALT), albumin, blood urea nitrogen (BUN), creatinine and estimated glomerular filtration rate (eGFR). Multivariate linear regression models were used to predict the mixed effect of the variables found to be potentially important from the bivariate analysis using *p* < 0.1 in point biserial correlations as a threshold to be included in the model.

## 3. Results

Abdominal MRI was available in 268 subjects with ADPKD including 127 males and 141 females, mean age = 47 ± 14 (range = 18–84) years (Figure 2, Table 2A). Compared with the control group, the ADPKD group was predominantly white, had a modestly higher diastolic BP and BUN, and a lower eGFR. As expected, ADPKD subjects had higher kidney, liver and spleen volumes. ADPKD subjects also had slightly higher AST and serum albumin levels compared with controls, although both were within normal ranges.

Indications for MRI in the control subjects included inflammatory bowel disease follow-up (*n* = 91), indeterminant abdominal finding requiring further investigation (*n* = 77), pain (*n* = 49), cancer with no active disease undergoing follow-up (*n* = 22), liver disease (*n* = 15), pancreatitis (*n* = 7), hematuria (*n* = 4), small bowel obstruction follow-up (*n* = 1), endometriosis (*n* = 1), and uterine leiomyomas (*n* = 1).

Matching CKD stage was possible for a subpopulation of 110 ADPKD and 110 control subjects to control for kidney function (Table 2B and Appendix A). In this subgroup, the mean eGFR was within the normal range in both ADPKD and control subjects (89 vs. 94 mL/min/1.73 m^2^, *p* = 0.2) and similar for the two groups. By contrast, the eGFR in the entire ADPKD group was lower than in the control group in Table 2 (70 vs. 92 mL/min/1.73 m^2^.

### 3.1. Interobserver Variability

There was good agreement among the 3 observers for identifying pleural fluid on MRI with kappa = 0.78. There was excellent agreement in the measurement of pleural fluid volume, ICC = 0.98.

### 3.2. Prevalence and Volume of Pleural Fluid in ADPKD and Control Groups

Dependent pleural fluid was identified on axial T2-weighted MR images in 56 of 268 (21%) ADPKD subjects (Table 2), compared with 21 of 268 controls (7.8%, *p* < 0.0001). For those ADPKD subjects with pleural effusion identified, the mean fluid volume was 20 ± 21 (range: 2–122) ml compared to 20 ± 27 (1–102) for control subjects with pleural effusions.

For the entire ADPKD group, eGFR was lower than in the control group (Table 2A, 70 vs. 92 mL/min/1.73 m^2^). To eliminate confounding due to a potential effect that this difference in eGFR may have a risk of pleural effusion occurrence, 110 ADPKD and 110 control subjects with similar eGFR were evaluated (Table 2B). In this analysis, the mean eGFR was within the normal range and similar for ADPKD and control groups (89 vs. 94 mL/min/1.73 m^2^, *p* = 0.2). As with the entire cohort of ADPKD and control groups, the ADPKD subjects in the GFR-matched subgroups also had a predominance of white sub-jects with slightly higher levels of BUN, albumin and AST, although they were all within the normal range.

Pleural effusion was identified on axial T2-weighted MR images in 28 of 110 (25%) ADPKD subjects, compared with 5 of 110 (5%) controls (*p* < 0.001). For this subgroup of ADPKD subjects with pleural effusion, the mean fluid volume was 21 ± 14 mL (range: 2–51 mL) compared to 18 ± 30 mL (range: 3–71 mL) for the control subgroup. This similarly higher prevalence of pleural effusion in ADPKD patients compared to controls for both the entire population and the sub-study controlling for eGFR supports the conclusion that the higher prevalence of pleural effusion in the ADPKD group was not due to CKD.

### 3.3. Genotype

Among 227 ADPKD patients with an identified pathogenic or likely pathogenic variant in either *PKD1* or *PKD2*, 38 of 171 (22%) with a *PKD1* mutation had pleural fluid compared to 9 of 56 (16%) with a *PKD2* mutation (*p* = 0.32) (Table 3). Mean pleural fluid volume was 18 ± 21 mL in PKD1 subjects and 28 ± 26 mL in PKD2 subjects (*p* = 0.32). For 227 subjects in whom PKD gene mutation analysis was available, 17 of 80 (21%) with truncating *PKD1* mutations had pleural fluid compared to 21 of 91 (23%) with non-truncating *PKD1* mutations (*p* = 0.77) and 5 of 34 (15%) with truncating *PKD2* mutations had pleural fluid compared to 4 of 22 (18%) with non-truncating PKD2 mutations (*p* = 0.73).

### 3.4. Correlation with Laboratory and Imaging Parameters

On bivariate regression, Table 4, the presence of pleural fluid on MRI of ADPKD subjects significantly correlated with female gender and negatively correlated with age, weight, body surface area, body mass index and ALT. For 112 male ADPKD subjects in whom imaging extended sufficiently into the pelvis to assess for free fluid, 26 had free pelvic fluid, which is not normally present in males. There was a weak correlation, r = 0.24 (*p* = 0.02), between free pelvic fluid and the presence of pleural effusions in men. For 87 female ADPKD subjects with imaging extending sufficiently into the pelvis to assess for free fluid, 72 had free pelvic fluid, which is normally expected before menopause, and female ADPKD subjects showed no significant correlation, r = 0.1 (*p* = 0.38), between free pelvic fluid and the presence of pleural effusion. There was no correlation of pleural fluid with height, blood pressure, liver volume, height-adjusted total kidney volume, spleen volume, AST, albumin, BUN creatinine, or eGFR. Only one subject was treated with tolvaptan at the time of MRI; that subject did not have a pleural effusion.

Multivariate analysis (Table 5) showed that age was the only parameter to retain statistical significance.

### 3.5. Clinical Effects and Duration of Pleural Effusion

None of the 56 ADPKD subjects with pleural effusion had any symptoms attributed to their pleural effusion. None had thoracentesis or other procedures to further characterize the pleural fluid [3,4,5,6,7,8]. The fluid signal was bright on T2-weighted images and dark on T1-weighted images in all patients suggesting simple fluid.

For 39 (70%) of all ADPKD subjects with pleural effusion, follow-up MRI was available ranging from 1.9 to 10.6 (mean = 5) years with a range of 2 to 7 scans (mean = 3.3) per subject. In these 39 ADPKD subjects with pleural effusions who had multiple MRIs, pleural effusion was identified on all follow-up MRIs in 35 of 39 (90%) (Figure 3). For the other 4 subjects, pleural effusion was identified on all but 1 follow-up scan in 2 and all but 2 follow-up scans in 2 subjects. For three ADPKD subjects with pleural effusion, CT was available within 2 months of the MRI. For all CTs the pleural effusion could be seen in retrospect, Figure 4, but was not mentioned in any of the CT reports.

## 4. Discussion

ADPKD is associated with abnormal fluid collections, including cysts in many organs, free pelvic fluid, dilated cisterna chyli, dural ectasia, pericardial effusions. In the current study of 268 ADPKD subjects and matched controls, we found that the pleural space is another location of increased fluid, with pleural effusion present in ~20% of ADPKD patients, approximately 3-fold greater than in the control group of patients without ADPKD. The multivariable model found a significant association of pleural effusion with younger age. Although pleural effusion can occur in severe chronic kidney disease, it was significantly more common in a subgroup of ADPKD subjects than in a control population matched for age, sex and renal function. Moreover, the high prevalence of pleural effusion occurred in ADPKD patients with well-preserved eGFR, further supporting the conclusion that severe CKD was not a significant factor in its pathogenesis.

There have been case reports of large pleural effusions in patients with ADPKD as well as in polycystic liver disease. These pleural effusions are often asymmetrical and have been attributed to cyst infections, lung cancer, cysto-pleural fistula, ascites with abdomino-pleural communications, cyst mass effect on diaphragm causing inflammation and decreased pleural capillary permeability and complicating hepatic cyst fenestration or resection [14,15,16,17,18,19,20,21,22]. Although none of our patients had large or symptomatic pleural effusions, the 20% prevalence we observed in ADPKD is much higher than expected based upon these few case reports in the literature.

Bronchiectasis has been reported to have high prevalence in ADPKD [10,23]. Drisco found decreased PKD1 expression in ADPKD lung compared to controls and hypothesized this leads to decreased ciliary function increasing the risk of bronchiectasis. There is also increased risk of nosocomial infections [11]. Both of these effects may contribute to pleural effusion but were not possible to study here because the patients did not undergo high resolution chest CT.

There are several cardiac changes in ADPKD that might contribute to the development of pleural effusions [24]. Mitral valve prolapse has been reported in 3.4 to 26 percent of patients and mitral regurgitation in 15.3% [25]. Hypertension is common in ADPKD, thought to be caused by cystic compression of renal parenchyma and vasculature [26]. Hypertension can lead to left ventricular hypertrophy and diastolic dysfunction. Diastolic dysfunction and mitral valve disease may contribute to increased pleural fluid by raising the pulmonary venous and intracardiac pressure.

Incidental, physiological pleural effusions have been reported in the general population. When imaging women prone, for breast MRI, Nyguen et al. found that pleural fluid could be identified by MRI in 174 of 200 (87%) of women [27]. This likely resulted from localization of the free pleural fluid in a small anterior space in the prone position, thereby increasing likelihood of its detection. Our subjects were all imaged supine, which distributes pleural fluid over a large area, making it more difficult to detect. Although the lateral chest X-ray is believed to detect pleural effusions as small as 5 or 10mL from blunting the costo-phrenic angle, decubitus X-rays were only able to detect pleural effusion in 1 subject out of a series of 106 healthy volunteers [28]. Ultrasound allowed positioning the subject to concentrate pleural fluid in a small pocket for easier visualization and was able to detect pleural effusion in 28 of 106 (26%) subjects [29]. These differences in rates of detecting pleural fluid are technique related and are controlled for in this study since both ADPKD and control subjects were imaged the same way.

The correlation of pleural effusion with free pelvic fluid in males, as well as with increased pericardial fluid and the association of dilated cisterna chyli with ADPKD raises the possibility of increased extracellular fluid volume due to an increase in total body fluid or third spacing due to impaired venous or lymphatic drainage [3,5]. However, the current study was not designed to address the mechanisms contributing to the development of pleural effusion in ADPKD.

None of the patients in this study had symptomatic pleural effusion, and thoracentesis was not performed, thus, we could not determine characteristics of the pleural fluid. However, the effusions appeared simple and dark on T1, suggesting a transudative process. Exudative effusions would be expected to appear more complex, with increased signal on T1 due to the paramagnetic effects of increased protein content.

The strengths of this study are the well characterized population of ADPKD subjects and the age and gender-matched control group, as well as a subgroup analysis of an eGFR-matched control group. Another strength of this study is the exclusion of subjects with heart failure, pulmonary disease and other potential clinical confounders. Limitations of this study include the relatively small sample size, retrospective design of the study, and the lack of thoracentesis data to further characterize the effusions. Although the abdominal MRIs did not cover the entire thorax, they did cover up to mid-thorax in all ADPKD and control subjects, which includes the portion of posterior pleural space where effusions accumulate in the supine position.

## 5. Conclusions

Pleural fluid is more prevalent in ADPKD subjects compared to control subjects without ADPKD. Although this did not appear to be associated with significant clinical complications, the increased prevalence of pleural effusion, together with earlier reports of an association of ADPKD with pericardial effusion, increased cisternae chyli diameter and pelvic fluid accumulation in males suggest that increased extravascular fluid is characteristic of this disorder, independent of estimated glomerular filtration rate. The potential role of the lymphatic system in the pathogenesis of ADPKD warrants consideration.

## Figures and Tables

**Figure 1 jcm-12-00386-f001:**
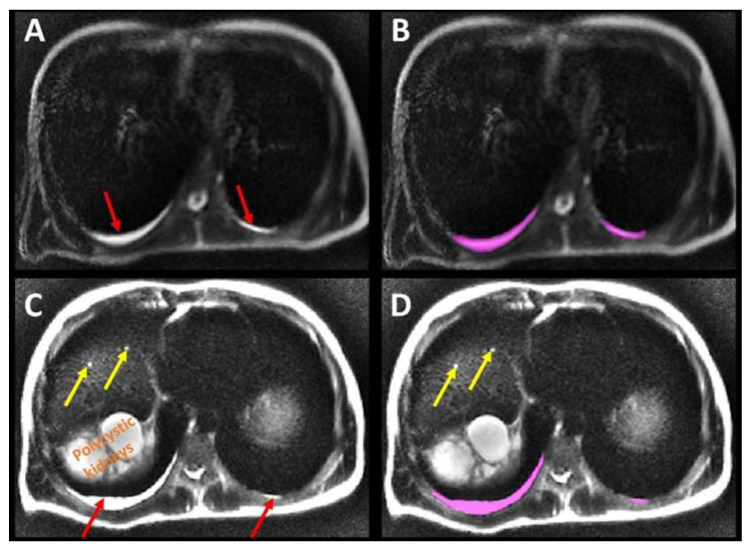
Axial T2 weighted images show pleural fluid (red arrows on (**A**,**C**)) which was measured by annotating the fluid, (**B**,**D**), in the thoracic cavity (pink label) to determine its volume. Liver cysts (yellow arrows) and the superior pole of the polycystic right kidney were also observed on the more inferior slices, (**C**,**D**).

**Figure 2 jcm-12-00386-f002:**
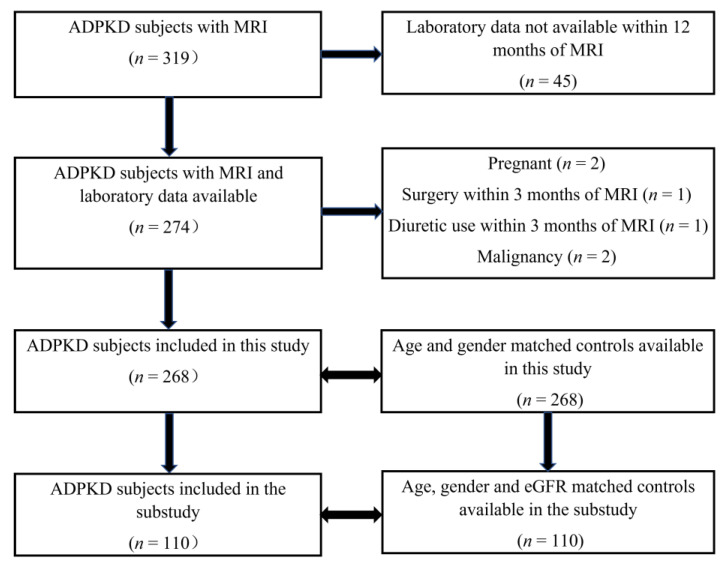
Patient flow chart. The “sub-study” corresponds to the cohort controlling for renal function (eGFR by CKD stage) in addition to age and gender.

**Figure 3 jcm-12-00386-f003:**
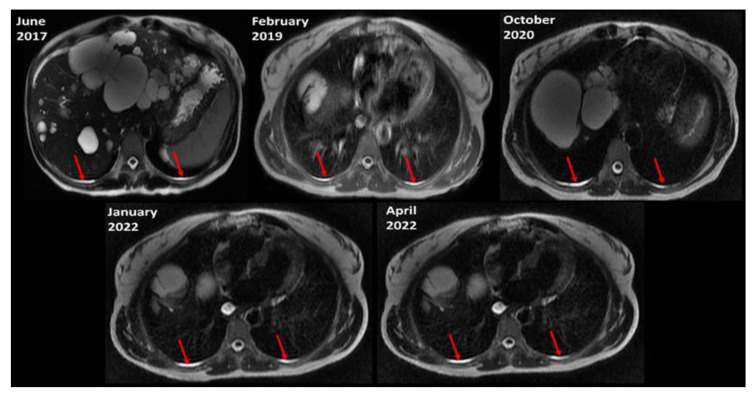
Pleural effusions (red arrows) are similar on successive axial T2-weighted MRI scans of the same subject over 5 years.

**Figure 4 jcm-12-00386-f004:**
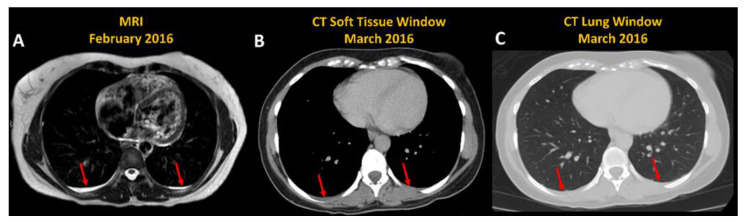
Pleural effusion (red arrows) seen on the same patient on (**A**) axial T2-weighted MRI and 1 month later on CT (**B**) soft tissue and (**C**) lung windows. Note that CT does not show the pleural fluid as well even though the amount of fluid is unchanged.

**Table 1 jcm-12-00386-t001:** MRI imaging parameters.

	Coronal SSFSE ^1^	Axial SSFSE	Axial 3D LAVA ^2^	Axial DWI ^3^	Axial SSFP ^4^
Image Weighting	T2	T2	T1	diffusion	T2/T1
Field of view	40–48	30–42	30–42	30–42	30–42
Matrix	320 × 256	320 × 208	288 × 192	160 × 108	192 × 244
Slice thickness	5 mm	5 mm	3 mm	5 mm	4 mm
TR ^5^/TE ^6^/flip	1200/91/130	1000/95/130	3.98/1.24/9	7500/90/90	3.9/1.7/60

^1^ SSFSE, single shot fast spin echo; ^2^ LAVA, liver imaging with volume acceleration; ^3^ DWI, diffusion-weighted imaging at B values = 50 and 800; ^4^ SSFP, steady-state free precession; ^5^ TR, repetition time; ^6^ TE, echo time.

**Table 2 jcm-12-00386-t002:** (**A**). Demographic and laboratory data in 268 ADPKD subjects and 268 controls matched for age and gender. Continuous variables are given as mean ± standard deviation. Ordinal variables are shown as number followed by the percentage. (**B**). Demographic and laboratory data in a subgroup of 110 ADPKD subjects and 110 matched controls who are also matched for renal function by CKD stage. Note that there is no difference in eGFR between the two subgroups but there continues to be a significantly higher prevalence of pleural effusion in ADPKD subjects.

(A)			
Demographic Data	ADPKD Subjects N = 268	Matched Controls N = 268	*p*-Value
Age	47 ± 14	47 ± 15	0.89
Male: Female (% Male)	127:141 (47%)	127:141 (47%)	1
** *Race* **			**<0.01 ***
**White**	**224 (84%)**	**173 (65%)**	
**Black**	**12 (4%)**	**29 (11%)**	
**Asian**	**20 (7%)**	**10 (4%)**	
**Native American**	**1 (1%)**	**0 (0%)**	
**Unknown**	**11 (4%)**	**56 (21%)**	
Weight (kg)	75 (18–146)	71 (37–156)	0.33
Body Mass Index (kg/m^2^)	25.3 (18.0–49.7)	25 (2–59)	0.90
Body Surface Area (m^2^)	1.9 (1.4–2.6)	0.8 (1.4–21.4)	0.22
Systolic Blood Pressure (mmHg)	120 (90–180)	120 (92–184)	0.43
**Diastolic Blood Pressure (mmHg)**	**79 ± 10**	**75 ± 9**	**<0.001 ***
**Estimated Glomerular Filtration Rate (mL/min/1.73 m²)**	**70 (5–132)**	**92 (24–150)**	**<0.001 ***
**Blood Urea Nitrogen (mg/dL)**	**20 (6–73)**	**12 (3–71)**	**<0.001 ***
**Albumin (g/dL)**	**4.3 (3.6–5.1)**	**4.1 (1.5–5.3)**	**<0.001 ***
**Aspartate Transaminase (U/L)**	**23 (15–113)**	**22 (11–125)**	**0.02 ***
Alanine Transaminase (U/L)	21 (10–138)	22 (6–153)	0.74
**Height-Adjusted Total Kidney Volume (mL/m)**	**782 (157–10,254)**	**190 (47–400)**	**<0.001 ***
**Liver Volume (mL)**	**1750 (854–15,563)**	**189 (47–400)**	**<0.001 ***
**Spleen Volume (mL)**	**236 (56–761)**	**182 (50–599)**	**<0.001 ***
**Pleural Fluid Present**	**56 (21%)**	**21 (8%)**	**<0.001 ***
Unilateral Right Pleural Fluid	2 (3.6%)	2 (9.5%)	0.33
Unilateral Left Pleural Fluid	2 (3.6%)	1 (4.8%)	0.82
**(B)**			
**Demographic Data**	**ADPKD Subjects** **N = 110**	**Matched Controls** **N = 110**	** *p* ** **-Value**
Age	44 ± 15	44 ± 15	1
Male: Female (% Male)	52:58 (53%)	52:58 (53%)	1
** *Race* **			**<0.01 ***
**White**	**92 (84%)**	**72 (65%)**	
**Black**	**6 (5%)**	**5 (5%)**	
**Asian**	**7 (6%)**	**3 (3%)**	
**Unknown**	**5 (4%)**	**30 (27%)**	
Weight (kg)	76 (61–87)	78 (63–87)	0.46
Body Mass Index (kg/m^2^)	26 (22–28)	26 (22–28)	0.47
Body Surface Area (m^2^)	1.9 (1.7–2.1)	1.9 (1.7–2.1)	0.85
*Blood Pressure*			
Systolic	122 (110–130)	122 (112–131)	0.63
Diastolic	78 ± 10	76 ± 8	0.15
Estimated Glomerular Filtration Rate (ml/min/173m^2^)	89 (67–110)	94 (77–115)	0.20
**Blood Urea Nitrogen (mg/dL)**	**18 (13–21)**	**15 (10–18)**	**<0.01 ***
**Albumin (g/dL)**	**4.4 (4.2–4.5)**	**4 (3.7–4.4)**	**<0.001 ***
**Aspartate Transaminase (U/L)**	**26 (20–28)**	**26 (18–28)**	**0.04 ***
Alanine Transaminase (U/L)	25 (17–28)	26 (15–28)	0.69
**Height-Adjusted Total Kidney Volume (mL/m)**	**781 (353–948)**	**196 (167–218)**	**<0.001 ***
Liver Volume	1973 (1403–2090)	1707 (1305–1889)	0.11
Spleen Volume	257 (173–306)	248 (164–287)	0.64
**Pleural Fluid Present**	**28 (25%)**	**5 (5%)**	**<0.001 ***

* indicates the difference is statistically significant.

**Table 3 jcm-12-00386-t003:** Demographic and laboratory data in 268 ADPKD subjects, with and without pleural fluid. Continuous variables are given as mean ± standard deviation. Ordinal variables are shown as number followed by the percentage.

Demographic Data	No Pleural Fluid N = 212	Pleural Fluid N = 56	*p*-Value
Age	48 ± 14	43 ± 14	0.007 *
Male: Female (% Male)	108:104 (51%)	19:37 (34%)	0.02 *
*Race*			0.07
White	182 (86%)	42 (75%)	
Black	8 (4%)	4 (7%)	
Asian	14 (7%)	10 (11%)	
Native American	0 (0%)	1 (2%)	
Unknown	8 (3%)	3 (5%)	
**Weight (kg)**	**70 (48–124)**	**77 (43–146)**	**0.02 ***
**Body Mass Index (kg/m^2^)**	**24 (17–49)**	**26 (18–44)**	**0.007 ***
**Body Surface Area (m^2^)**	**1.8 ± 0.2**	**1.9 (1.4–2.6)**	**0.03 ***
*Blood Pressure*			
Systolic	121 (94–171)	120 (90–180)	0.69
Diastolic	79 ± 8	79 ± 10	0.88
Estimated Glomerular Filtration Rate (mL/min/1.73 m²)	83 (5–130)	69 (6–132)	0.30
Blood Urea Nitrogen (mg/dL)	18 (6–57)	20 (6–73)	0.14
Albumin (g/dL)	4.3 (3.6–4.8)	4.3 (3.6–5.1)	0.22
Aspartate Transaminase (U/L)	23 (15–37)	23 (15–113)	0.28
**Alanine Transaminase (U/L)**	**18 (10–44)**	**21 (10–138)**	**0.02 ***
Total Kidney Volume/height (mL/m)	653 (216–10,254)	785 (153–6458)	0.73
Liver Volume (mL)	1707 (1054–15,563)	1762 (854–7589)	0.55
Spleen Volume (mL)	247 (117–533)	234 (56–761)	0.34
*Genotype Data*			
Data Available	180 (85%)	47 (84%)	0.86
PKD1 Mutations	133 (74%)	38 (81%)	0.32
PKD 2 Mutation	47 (26%)	9 (19%)	0.32

* indicates the difference is statistically significant.

**Table 4 jcm-12-00386-t004:** Bivariate correlations between pleural effusion and clinical/laboratory variables in ADPKD subjects.

Variables	Correlation Coefficient	*p*-Value
Age	−0.15	0.012 *
**Gender (female =0, male = 1)**	**−0.14**	**0.023 ***
Height	−0.06	0.336
**Weight**	**−0.14**	**0.021 ***
**Body Mass Index**	**−0.12**	**0.045 ***
**Body Surface Area**	**−0.13**	**0.032 ***
*Blood Pressure*		
Systolic	−0.02	0.747
Diastolic	−0.002	0.972
Estimated Glomerular Filtration Rate	0.07	0.281
Blood Urea Nitrogen	−0.017	0.227
Albumin	−0.09	0.129
Aspartate Transaminase	−0.08	0.183
**Alanine Transaminase**	**−0.13**	**0.038 ***
PKD Mutation Genotype	0.01	0.814
Total Kidney Volume/height	0.02	0.744
Liver Volume	0.03	0.640
Spleen Volume	0.03	0.606

* indicates the difference is statistically significant.

**Table 5 jcm-12-00386-t005:** Multivariate analysis on parameters with *p* < 0.1 from bivariate analysis (model *p*-value = 0.014).

Demographic Data	Coefficients	Standard Error	*t*-Stat	*p*-Value	Lower 95%	Upper 95%
Intercept	0.62	0.14	4.36	0.00002	0.34	0.90
**Age**	**−0.004**	**0.002**	**−2.32**	**0.02 ***	**−0.008**	**−0.0006**
Gender	−0.07	0.07	−0.91	0.36	−0.21	0.08
Weight	−0.0008	0.004	−0.21	0.84	−0.008	0.007
Body Mass Index	−0.002	0.01	−0.18	0.86	−0.03	0.02
Alanine Transaminase	−0.003	0.002	−1.28	0.20	−0.007	0.002

* indicates the difference is statistically significant.

## Data Availability

Data generated or analyzed during this study are available from the corresponding author by request subject to institutional review and a data use agreement.

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
