# Peer review of "Pleural Effusions on MRI in Autosomal Dominant Polycystic Kidney Disease"

_jcm, 2023, doi:10.3390/jcm12010386_

Round 1

Reviewer 1 Report

In this study by Dr. Martin R. Prince and colleagues, the authors found that pleural fluid is more prevalent in ADPKD subjects compared to control groups without ADPKD. This is an interesting study; however, I have several concerns below,

1.     Based on the matched eGFR data, the authors claimed that the higher prevalence of pleural effusion in the ADPKD group was not due to CKD. I am so curious about another variable in CKD, Cystatin C..Is there any further data about cystatin C?

2.     Is there any further evidence about pleural effusion in right or left?

3.     Proof-reading for the language is needed, including the grammatical errors and some spelling errors. Also, terms should be defined the first time they are mentioned.

Author Response

1.     Based on the matched eGFR data, the authors claimed that the higher prevalence of pleural effusion in the ADPKD group was not due to CKD. I am so curious about another variable in CKD, Cystatin C..Is there any further data about cystatin C?  We now clarify that cystatin C was not available in these patients because it was not part of the study protocol.  (see page 3, Line 87). 

2.     Is there any further evidence about pleural effusion in right or left?  We now indicate how many patients had unilateral pleural effusion on the right and on the left, (see Table 2A, last 2 lines). It was only a few subjects who had unilateral pleural effusions.

3.  Proof-reading for the language is needed, including the grammatical errors and some spelling errors. Also, terms should be defined the first time they are mentioned.  The manuscript has now been thoroughly proofread by a native English speaker correcting all grammatical and spelling errors.

Reviewer 2 Report

The manuscript Pleural Effusions on MRI in Autosomal Dominant Polycystic 2 Kidney Disease by Jin Liu, Xiaorui Yin, Hreedi Dev, Xianfu Luo, Jon D. Blumenfeld, Hanna Rennert, and Martin R. Prince describes the existence of a potential relationship of Autosomal dominant polycystic kidney disease ( ADPKD) with pleural effusion. The subject of the presented manuscript is very interesting and will provide interesting and clinically useful data. The construction of the manuscript is also typical and appropriate, with the manuscript itself requiring a few minor corrections, which I present below.

General tips

1. Dear authors, please check the entire manuscript for editorial and spacing before citation numbers, e.g. lines 41, 43, 44. Please check the entire manuscript.

Tabels:

1. In line 76, there is a reference to table 1, unfortunately I do not see it in the text, is the table numbering in the manuscript correct?

2. Please check the journal guidelines for the preparation of tables in the manuscript and then adapt the tables in the text to the guidelines.

3. In my opinion, it would be clearer to mark the statistical results in bold, but also to add the "*" symbol. The symbol should be explained

Discussion:

1. The topic of the presented research is niche, but I would like to ask you to deepen the topic and search the pubmed and scopus databases again in order to find articles that deepen the subject and broaden the discussion as much as possible

Author Response

  1. Dear authors, please check the entire manuscript for editorial and spacing before citation numbers, e.g. lines 41, 43, 44. Please check the entire manuscript. done

Tables: 

  1. In line 76, there is a reference to table 1, unfortunately I do not see it in the text, is the table numbering in the manuscript correct?  Table 1 was mistakenly omitted and is now included.
  2. Please check the journal guidelines for the preparation of tables in the manuscript and then adapt the tables in the text to the guidelines.  All tables have been adjusted and corrected to meet the guidelines.
  3. In my opinion, it would be clearer to mark the statistical results in bold, but also to add the "*" symbol. The symbol should be explained In the Tables, statistically significant results are now BOLD and marked with a “*”.

Discussion: 

  1. The topic of the presented research is niche, but I would like to ask you to deepen the topic and search the pubmed and scopus databases again in order to find articles that deepen the subject and broaden the discussion as much as possible    Additional papers including two important JCM papers on ADPKD are now referenced (see references 10-12 and 23-29) to deepen and broaden the introduction and discussion.

Reviewer 3 Report

The authors retrospectivelly evaluated the presence of pleural effusions on abdominal MRI in ADPKD patients and matched controls. I find it unfortunately has major limitations:

- The introduction and discussion section are very short and do not highlight the scientific interest of this study.

- No information is given about how the matching of the controls was done. Furthermore, the authors claim that the controls were matched on eGFR, but the eGFR is different between the ADPKD and control groups. ADPKD patients also had higher blood pressure, which can be an important confounder when evaluating fluid accumulation.

- Left ventricular ejection fraction was not evaluated is these patients, which can also be a major confounder.

- The volume of pleural effusion described is small. Yet, it is know that small pleural effusions can be physiological (Nguyen J et al., Am J Roentgenol 2012 / Kocijancic I et al., Clin Radiol 2004)

- The protocol codes of the IRB seem ancient and are the same that a similar article about pericardial effusion in ADPKD recently published by the authors (Liu J et al. J Clin Med 2022).

Author Response

- The introduction and discussion section are very short and do not highlight the scientific interest of this study.  The introduction and discussion sections have been expanded to highlight the scientific interest in this study see page 2 – introduction and pages 10-11, discussion. 

- No information is given about how the matching of the controls was done. Furthermore, the authors claim that the controls were matched on eGFR, but the eGFR is different between the ADPKD and control groups. ADPKD patients also had higher blood pressure, which can be an important confounder when evaluating fluid accumulation.  Further details are now provided about the matching of controls in the sub-study which was done based upon CKD stage (page 3, lines 78-80). Actually the eGFR was not different in the group where eGFR was matched see Table 2B (previously as Supplemental Table 1).  To highlight this point, we have moved the previous Supplemental Table 1 into the manuscript as Table 2B. 

- Left ventricular ejection fraction was not evaluated is these patients, which can also be a major confounder.  We thank reviewer 3 for making this important point because cardiac disease is a potential confounder.  A strength of this study is that we excluded patients with heart failure and other potential clinical confounders,  although we could have overlooked patients with sub-clinical disease. This is now mentioned in the discussion, see pages 11 and 12.  

- The volume of pleural effusion described is small. Yet, it is know that small pleural effusions can be physiological (Nguyen J et al., Am J Roentgenol 2012 / Kocijancic I et al., Clin Radiol 2004).  It is now pointed out that these small pleural effusions might not have any pathological significance other than correlating with ADPKD.  These papers are now referenced and their methodological differences including prone or decubitus positioning which concentrates pleural fluid into a smaller but thicker pocket which is easier to visualize is explained (see discussion, line 286-298 on page 11). 

- The protocol codes of the IRB seem ancient and are the same that a similar article about pericardial effusion in ADPKD recently published by the authors (Liu J et al. J Clin Med 2022).  It is now clarified that these papers both come from the same research repository study at the Rogosin Institute (see page 3, lines 80-81). 

Round 2

Reviewer 1 Report

the terms should be defined the first time they are mentioned (such as Pkd1)..I would like to ask the authors to proofread the whole manuscript thoroughly, although the authors claimed that it has been proofread.

Author Response

the terms should be defined the first time they are mentioned (such as Pkd1).

PKD1, PKD2 and several other terms that were not previously defined (including MRI, HIPAA, ANOVA) and now defined the first time they are mentioned.

I would like to ask the authors to proofread the whole manuscript thoroughly, although the authors claimed that it has been proofread.  The authors thank reviewer 1 for this additional opportunity to thoroughly proofread the manuscript.

Reviewer 3 Report

I thank the authors for adressing my concerns. The manuscript is improved and clearer, however I still have one remark:

- It is still unclear how the matching was done in Table 2A. The text state that it was done on eGFR, but eGFR is different. Please correct it and give more details on how you selected the patients in the control group to match the ADPKD population.

Author Response

It is still unclear how the matching was done in Table 2A. The text state that it was done on eGFR, but eGFR is different. Please correct it and give more details on how you selected the patients in the control group to match the ADPKD population.  We now clarify that that the subgroup with renal function matching by CKD stage in addition to age and gender is shown in Table 2B (see page 6, lines 158-166).  The title of Table 2A is now clarified to indicate that Table 2A is matched for age and gender.